# Roles of the Arabidopsis *KEULE* Gene in Postembryonic Development

**DOI:** 10.3390/ijms25126667

**Published:** 2024-06-18

**Authors:** Alejandro Ruiz-Bayón, Carolina Cara-Rodríguez, Raquel Sarmiento-Mañús, Rafael Muñoz-Viana, Francisca M. Lozano, María Rosa Ponce, José Luis Micol

**Affiliations:** Instituto de Bioingeniería, Universidad Miguel Hernández, Campus de Elche, 03202 Elche, Spain; alejandro.ruizb@umh.es (A.R.-B.); 22car0lina5@gmail.com (C.C.-R.); rsarmiento@umh.es (R.S.-M.); rafael.munoz.viana@gmail.com (R.M.-V.); fmlozanogarcia@gmail.com (F.M.L.); mrponce@umh.es (M.R.P.)

**Keywords:** Arabidopsis, *KEULE* gene, SNARE, SM proteins, vesicle fusion, leaf margin patterning

## Abstract

Cytokinesis in plant cells begins with the fusion of vesicles that transport cell wall materials to the center of the cell division plane, where the cell plate forms and expands radially until it fuses with the parental cell wall. Vesicle fusion is facilitated by *trans*-SNARE complexes, with assistance from Sec1/Munc18 (SM) proteins. The SNARE protein KNOLLE and the SM protein KEULE are required for membrane fusion at the cell plate. Due to the crucial function of KEULE, all Arabidopsis (*Arabidopsis thaliana*) *keule* mutants identified to date are seedling lethal. Here, we identified the Arabidopsis *serrata4-1* (*sea4-1*) and *sea4-2* mutants, which carry recessive, hypomorphic alleles of *KEULE*. Homozygous *sea4-1* and *sea4-2* plants are viable and fertile but have smaller rosettes and fewer leaves at bolting than the wild type. Their leaves are serrated, small, and wavy, with a complex venation pattern. The mutant leaves also develop necrotic patches and undergo premature senescence. RNA-seq revealed transcriptome changes likely leading to reduced cell wall integrity and an increase in the unfolded protein response. These findings shed light on the roles of KEULE in postembryonic development, particularly in the patterning of rosette leaves and leaf margins.

## 1. Introduction

During cytokinesis, the cytoplasm of a cell is divided, forming two daughter cells. In plant cells, the phragmoplast, a dynamic matrix of antiparallel microtubules, transports vesicles containing cell wall materials from the *trans*-Golgi network to the center of the cell division plane. These vesicles accumulate in the cell division plane and fuse together, forming the cell plate, which expands radially toward the plasma membrane of the parental cell. Fusion between the cell plate and plasma membrane occurs at the cortical division site where the preprophase band, a densely packed matrix of actin filaments and antiparallel microtubules, was formed. Finally, callose and cellulose synthases modify the newly formed cell wall, increasing its rigidity and physically separating the daughter cells [1].

Proteins involved in cytokinesis in *Arabidopsis thaliana* (hereafter Arabidopsis) are classified into three categories: proteins responsible for the proper orientation of the division plane, proteins directly involved in executing cytokinesis, and proteins involved in cell wall biosynthesis [2]. The first category includes proteins such as TONNEAU1 (TON1) and TON2 [3], TON1-RECRUITING MOTIF1 (TRM1; [4]), and TANGLED1 (TAN1; [5]). The second category comprises proteins that organize antiparallel microtubules, such as the kinesin RADIALLY SWOLLEN7 (RSW7; [6]) and the tubulin folding cofactors PILZ [7]; proteins involved in phragmoplast reorganization, such as the microtubule-associated kinase RUNKEL (RUK; [8]), the kinesin HINKEL (HIK; [9]) and the microtubule-associated protein PLEIADE (PLE; [10]); and proteins that participate in vesicle fusion at the cell plate, such as the subunits of the Transport Protein Particle II complex (TRAPPII), the tethering factor CLUB [11], the syntaxin KNOLLE (KN; [12]), and the Sec1/Munc18 (SM) protein KEULE (KEU; [13]). The third category includes callose synthases such as MASSUE (MAS; [2]) and CALLOSE SYNTHASE1 (CALS1; [14]), cellulose synthases such as PROSCUTE1 (PRC1; [15]) and RADIALLY SWOLLEN1 (RSW1; [16]), the GDP-mannose pyrophosphorylase CYTOKINESIS DEFECTIVE1 (CYT1; [17]), and the endo-1,4-beta-D-glucanase KORRIGAN (KOR; [18]).

The formation of the cell plate during cytokinesis occurs through the fusion of vesicles from the *trans*-Golgi network. Trafficking of these vesicles begins during anaphase, when the phragmoplast transports the vesicles to the cell division plane along with other membrane and cell wall components. In Arabidopsis, the Soluble N-ethylmaleimide-Sensitive Factor (NSF) Attachment Protein Receptor complexes (SNARE) play a central role in vesicle fusion [19]. These complexes, along with other components of the vesicle fusion machinery, are highly conserved in eukaryotic cells and have been studied in many species [20].

Vesicle fusion requires proper tethering and docking [21]. During tethering, specific proteins on a vesicle surface capture other vesicles, keeping them close to facilitate their fusion [22]. These tethering proteins have supercoiled helical structures or are part of protein complexes such as TRAPP or the exocyst [21]. During docking, SNARE complexes form between a vesicle and a cell membrane (heterotypic fusion) or between biochemically identical membranes (homotypic fusion), which brings the merging membranes closer to each other [23]. SNARE complexes consist of four SNARE domains, each containing an alpha-helix anchored to a membrane via its C-terminus. SNARE components anchored to a vesicle are referred to as v-SNARE or R-SNARE, as they possess an arginine (R) residue at the center of the SNARE domain. SNARE components anchored to a target membrane are known as t-SNARE or Q-SNARE, as they contain a glutamic acid (Q) residue at the same position. The SNARE complex consists of four components: a SNARE domain from an R-SNARE, another from a syntaxin (Qa-SNARE domain), and two domains from Synaptosomal-Associated Protein 25 (SNAP25; Qbc-SNARE domain) or from two independent proteins (Qb-SNARE and Qc-SNARE domains) [24].

Syntaxins can adopt closed or open conformations. In the closed conformation, the Habc N-terminal domain, consisting of three alpha-helices, folds over its SNARE domain. In the open conformation, the Habc and SNARE domains remain separate [25]. The syntaxin KNOLLE plays a central role in cytokinesis. In Arabidopsis, KNOLLE is transported by *trans*-Golgi vesicles in *cis*-SNARE complexes with the Qbc-SNARE SOLUBLE N-ETHYLMALEIMIDE-SENSITIVE FACTOR ADAPTOR PROTEIN33 (AtSNAP33), which is the Arabidopsis homolog of human SNAP25 [26], and the R-SNARE VESICLE-ASSOCIATED MEMBRANE PROTEIN721 (VAMP721). At the division plane, the ATPase NSF separates the components of the *cis*-SNARE complexes, and KNOLLE adopts a closed conformation, preventing its interaction with other SNARE components. During cytokinesis, the presence of KEULE stabilizes the open conformation of KNOLLE by interacting with the linker region between its domains [23]. This stabilization promotes the formation of *trans*-SNARE complexes with the SNARE components of adjacent vesicles [23]. KEULE also interacts with other syntaxins in heterotypic fusions, such as the closed conformation of SYNTAXIN-RELATED PROTEIN1 (SYP121), which participates in vesicle trafficking of the cell membrane during drought stress and pathogen defense [27].

All Arabidopsis *keule* (*keu*) mutants studied to date are seedling lethal and have large multi-nucleated cells with incomplete cell walls in dividing embryonic cell populations [28]. Seedlings with the same *keu* genotype can exhibit various morphologies, ranging from balls of undifferentiated cells to seedlings with well-defined organs, although most are rod-shaped [28]. Seedlings with the most severe mutant phenotype exhibit a swollen epidermis, with poorly defined cellular layers containing undifferentiated cells in both the epidermis and inner tissues. This is not observed in seedlings with milder morphologies. The division planes in the *keu* mutants are improperly oriented, and cytokinesis occurs at a slower pace [28]. During embryogenesis, the division of the smallest daughter cell of the zygote occurs perpendicular instead of parallel to the apical–basal axis. Additionally, the asynchronous tangential divisions of the zygote often lead to the partial or total loss of the protoderm. However, these embryos usually recover this aspect of the wild-type phenotype during development [28].

Here, we describe two new hypomorphic alleles of the Arabidopsis *KEU* gene, *serrata4-1* (*sea4-1*) and *sea4-2*, which are viable and fertile. The *sea4-1* mutation causes retention of the ninth intron of *KEU* and the *sea4-2* mutation is predicted to induce an amino acid substitution in the SNARE interaction domain of KEU. Both mutants display reduced rosette size and plant height, abnormal leaf venation patterns, variable cell sizes within the epidermis and palisade mesophyll, and apparently undifferentiated cells throughout these layers, as well as premature leaf senescence. Our findings shed light on the functions of *KEULE* in adult plants, paving the way for future investigation of the postembryonic roles of this essential gene.

## 2. Results

### 2.1. The sea4 Mutants Carry Novel Hypomorphic Viable Alleles of KEU

In a large-scale screening for Arabidopsis leaf mutants conducted after ethyl methanesulfonate (EMS) mutagenesis of Landsberg *erecta* (L*er*) seeds [29], we isolated two mutants displaying serrated and wavy rosette leaves with necrotic patches. These mutants were found to be allelic and were named *serrata4-1* (*sea4-1*) and *sea4-2*. Initially, the *sea4-1* and *sea4-2* mutations were mapped at low resolution to a 660 kb candidate region of chromosome 1 containing approximately 250 genes (Figure 1A) using iterative linkage analysis to molecular markers (Appendix A; [30]). We generated backcross mapping populations by selecting F_2_ plants exhibiting the mutant phenotype and subjected their pooled genomic DNA to next-generation sequencing. We analyzed the resulting paired-end reads using Easymap v.2, which also pointed to the same region of chromosome 1 (Figure 1B). By comparing the lists of candidate mutations identified in *sea4-1* (Appendix A) and *sea4-2* (Appendix A), we identified At1g12360 as the most likely candidate gene, which encodes the Sec1/Munc18 protein KEULE (KEU), as both mutants carried a mutation in this gene.

We obtained three lines from the SALK collection that were annotated to carry T-DNA insertions within the At1g12360 transcription unit: SALK_101874C, SALKseq_085463, and SALKseq_089213. We renamed the mutations in these lines *keu-21*, *keu-22*, and *keu-23*, respectively (Figure 1C). Additionally, Prof. Gerd Jürgens provided us with the line *keu^MM125^* [31], which carries a 100 bp deletion in the 16th exon of *KEU* (Figure 1C). As expected, plants homozygous for these putatively null *keu* alleles were embryonic lethal, and the rosettes of the *KEU/keu-21*, *KEU/keu-22*, *KEU/keu-23*, and *KEU/keu^MM125^* heterozygotes were phenotypically wild type. Crosses of *KEU/keu-21*, *KEU/keu-22*, and *KEU/keu-23* heterozygotes to homozygous *sea4/sea4* plants produced viable heterozygous *sea4/keu* plants, exhibiting a mutant phenotype more severe than that of *sea4-1/sea4-1* or *sea4-2/sea4-2* homozygotes (Figure 2). Crosses of *KEU/keu^MM125^* heterozygotes to homozygous *sea4/sea4* plants produced heterozygous *sea4/keu^MM125^* plants displaying a similar phenotype (Figure 2), but they died at the end of the vegetative phase without bolting or producing inflorescences. These results confirm the notion that *sea4-1* and *sea4-2* are hypomorphic alleles of *KEU*.

### 2.2. The sea4-1 Mutation Causes Mis-Splicing of the KEU Pre-mRNA, and sea4-2 Appears to Perturb the Secondary Structure of the KEU Protein

The Arabidopsis *KEU* gene is expressed throughout the plant, especially in tissues undergoing division. KEU exists in soluble form in the cytoplasm or associates with membranes during cytokinesis. KEU is involved in cytokinesis, but not in cellular elongation. KEU shares 28−30% identity with its Sec1 orthologs in mammals, *Caenorhabditis elegans*, and *Drosophila melanogaster*, and 61% and 65% identity with its Arabidopsis homologs SEC1A and SEC1B, respectively [13].

In *sea4-1*, a G→A transition at the 3′ end of the ninth exon of *KEU* damages the splice donor site. In *sea4-2*, a C→T transition in the third exon is predicted to result in an S57L substitution (Figure 1C). Since *sea4-1* is likely to undergo mis-splicing and produce abnormal mRNA and protein products, we conducted RNA-seq analysis of this line, finding that 99.17% of its mature mRNAs retained the ninth intron (Appendix A), leading to a premature stop codon. The mutant SEA4-1 protein consists of 302 amino acids, 38 of which are absent from wild-type KEU (Appendix A). The expression level of *KEU* was five times higher in the *sea4-1* mutant than in L*er*, as indicated by the number of RNA-seq reads aligned to *KEU*, likely to compensate for the near absence of wild-type mRNAs (Appendix A).

Alignment of the KEU sequence with its rat STXBP1 ortholog revealed that the *sea4-2* mutation alters one of the amino acids that physically interact with syntaxins (Appendix A; [32]). To assess the impact of this mutation on the structural stability and dynamics of KEU, we used Dynamut and DynaMut2 to predict differences in the unfolding Gibbs free energy (ΔΔG) and vibrational entropy energy (ΔΔS_Vib_) between the wild-type and mutant proteins. We obtained the structure of the full-length KEU protein from the AlphaFold Protein Structure Database. However, these two predictors provided conflicting ΔΔG values, suggesting that the S57L substitution does not affect protein stability (Appendix A). ENCoM calculated a decrease in flexibility of KEU due to this mutation based on the ΔΔS_Vib_ value (Appendix A). To predict the effects of the S57L substitution on KEU protein structure, we used Missense 3D. This software did not predict any structural damage to KEU, since both residues were exposed to the solvent in a similar manner. However, two hydrogen bonds between S57 and its neighboring residues V53 and K54 were lost (Appendix A), and the cavity volume increased by 24.408 Å^3^.

### 2.3. Morphological and Histological Characterization of the sea4 Mutants

#### 2.3.1. The sea4 Mutants Exhibit a Pleiotropic Morphological Phenotype

Both *sea4* mutants exhibited wavy leaves with serrated margins and senescent patches (Figure 2). Both *sea4-1* and *sea4-2* displayed phenotypic variability, including varying rates of cotyledon and leaf expansion and variations in the extent of senescent patches (Appendix A). A similar phenotypic variability was observed in lethal seedlings carrying other *keu* mutant alleles [28] and reappeared in the progeny of selfed *sea4* plants, regardless of the parental phenotype (Appendix A). Inflorescences of *sea4* plants contained fertile flowers that opened prematurely before complete maturation (Appendix A). Inflorescences of *KEU/keu-21*, *KEU/keu-22*, and *KEU/keu-23* plants were indistinguishable from those of Col-0 (Appendix A). However, in inflorescences of heterozygotes of *sea4-1* or *sea4-2* with *keu21*, *keu22*, or *keu23*, only a few flowers opened prematurely. These flowers exhibited short sepals that were separated from each other and had protuberances on their margins, but they remained fertile (Appendix A). Inflorescences of *KEU/keu^MM125^* plants produced a few fertile flowers that began to open slightly before reaching complete maturation (Appendix A), while heterozygous *sea4/keu^MM125^* plants did not produce inflorescences, as already mentioned. Taken together, the latter two observations suggest that *keu^MM125^* is not null; rather, it is an antimorphic allele. Inflorescences of *sea4-1/sea4-2* plants contained fewer flowers than those of homozygous *sea4* mutants, but they exhibited the same phenotypes (Appendix A). The height of *sea4-1*, *sea4-2*, and *KEU/keu^MM125^* plants was reduced compared to L*er*, whereas *KEU/keu-21*, *KEU/keu-22*, and *KEU/keu-23* plants did not exhibit a significant difference from the wild type. *sea4-1/sea4-2* plants displayed an intermediate height between that of *sea4-1* and *sea4-2* homozygous plants. Heterozygous *sea4/keu* plants were even smaller, and individuals carrying a *sea4-1* allele were smaller than those carrying a *sea4-2* allele, which is consistent with the smaller height observed in *sea4-1* vs. *sea4-2* plants (Appendix A). The primary root length was also reduced in the *sea4* mutants compared to the wild type (Appendix A).

#### 2.3.2. The sea4 Leaves Have a Dense and Complex Venation Pattern

The presence of serrations in *sea4* leaves pointed to the existence of other alterations in the internal structures of these leaves. To observe possible alterations in the venation pattern, we decolorized cotyledons, first-node leaves, and third-node leaves of *sea4-1* (n = 12) and *sea4-2* (n = 13–15) plants and compared their venation patterns to L*er* (n = 12–15) (Figure 3). The *sea4-1* cotyledons showed an increased number of terminal veins per unit area compared to L*er*, whereas *sea4-2* cotyledons were smaller and exhibited increased vein length and bifurcation number per unit area. The first-node leaves of *sea4-1* and *sea4-2* were elongated, smaller, and contained more terminal veins per unit area than L*er*. Additionally, *sea4-1* first-node leaves displayed increased vein length and bifurcations per unit area compared to L*er*. The third-node leaves of *sea4-1* and *sea4-2* were smaller, with an increase in terminal veins per unit area compared to L*er*. In *sea4-2*, the leaves were also elongated and exhibited increased vein length and bifurcations per unit area compared to L*er* (Appendix A). Overall, the cotyledons and leaves of the *sea4* mutants were smaller and elongated compared to the wild type. They showed a higher vein density, and their venation patterns were more complex than wild type, with a greater number of bifurcations and terminal veins per unit area.

#### 2.3.3. sea4 Leaves Show Aberrant Epidermal and Mesophyll Structure

We obtained transverse sections of third-node leaves from *sea4-1*, *sea4-2*, and L*er* plants (Figure 4). In the palisade mesophyll of the *sea4* mutants, the cells were generally well organized but were more variable in size compared to L*er*. Some large cells were interspersed among small cells within the palisade mesophyll. The spongy mesophyll in the *sea4* mutants was disorganized, with very large cells surrounded by small cells arranged in multiple layers. The epidermal cells appeared mostly normal, although protuberances were observed on the surfaces of mutant leaves. These protuberances appeared to result from increased local proliferation or growth of mesophyll cells.

We quantified the variation in palisade mesophyll, adaxial epidermis, and abaxial epidermis cells in transverse sections of third- and first-node leaves (Figure 5 and Appendix A). In the palisade mesophyll of the two leaves analyzed, most *sea4* cells were smaller than L*er* cells, but a small fraction of *sea4* cells were larger. In the epidermis, cells tended to have a similar size, although *sea4* cells had a simpler shape compared to L*er* cells, with fewer protuberances observed in both first- and third-node leaves.

#### 2.3.4. The sea4 Mutants Show Early Leaf Senescence and Bolting

A characteristic trait of the *sea4* mutant leaves is the early appearance of senescent patches, which gradually expand until the leaves are completely senescent. To quantify the degree of senescence, we grouped all leaves from *sea4-1* (n = 38), *sea4-2* (n = 45), and L*er* (n = 44) plants into five phenotypic classes based on their degree of senescence at 25 days after stratification (das; Appendix A). L*er* plants displayed some senescence in their first four leaves, particularly in the first and second nodes (Appendix A). *sea4-1* leaves exhibited senescence from the first to eighth nodes (Appendix A), with all leaves showing more severe senescence compared to L*er*. Similarly, *sea4-2* plants displayed senescence in their first seven leaves (Appendix A), with leaves in the first and second nodes exhibiting more severe senescence than L*er*. In conclusion, *sea4* plants demonstrated a more pronounced state of senescence in their initial leaves compared to the wild type and exhibited senescence starting from the fifth-node leaves onwards, which was not observed in L*er*.

Senescence appeared earlier in *sea4* than in the wild type. The first signs of senescence appeared ~14 das in *sea4-1* (n = 95) and *sea4-2* (n = 96) plants and ~21 das in L*er* (n = 79; Appendix A). The early onset of senescence was consistent across different phenotypic classes in the *sea4* mutants (Appendix A) and occurred at the same time in both *sea4-1* and *sea4-2* (Appendix A). Only *sea4-2* exhibited a significant reduction in bolting time compared to L*er* (Appendix A). When considering the phenotypic classes of the *sea4* mutants, there was a positive correlation between leaf mutant phenotype severity and an increase in bolting time (Appendix A).

### 2.4. Double Mutant Combinations of sea4 Alleles and Alleles of Genes Encoding Proteins Required for Membrane Fusion Show Defects in Leaf Development

As the *sea4* mutants carry the first known viable alleles of *KEU*, we combined them with alleles of other genes involved in membrane fusion to characterize their genetic interactions in adult plants. *SEC1B* is a *KEU* homolog in Arabidopsis that is expressed at very low levels, but its overexpression rescued the phenotype of *keu* mutants [31]. We obtained two lines, *sec1b-1* and *sec1b-2*, carrying T-DNA insertions interrupting *SEC1B* in the twelfth intron and first exon, respectively (Appendix A), and showing a wild-type phenotype as homozygotes (Figure 6E,F). The *sea4 sec1b-1* double mutants displayed small rosettes with small, highly serrated leaves. The *sea4 sec1b-2* double mutants were very small, exhibited fully expanded cotyledons, and lacked true leaves, instead showing small protuberances emerging from the shoot apical meristem (Figure 6L–O).

*SEC6* encodes a protein that is part of the exocyst complex, which colocalizes with KEU in the cellular plate [33]. We also obtained the lines *sec6-2* and *sec6-3*, carrying T-DNA insertions in the 23rd and 26th exons of *SEC6*, respectively (Appendix A), which presented a wild-type phenotype as homozygotes (Figure 6G,H). The *sea4-1 sec6-2* and *sea4-1 sec6-3* double mutants displayed a reduced size with small, wavy, heavily serrated leaves, and *sea4-2 sec6-2* and *sea4-2 sec6-3* exhibited a small size, with their leaves showing minimal or no development (Figure 6P–S).

We also crossed the *sea4* mutants with lines carrying mutations in genes encoding syntaxins: *SYP21*, *SYP132*, and *KNOLLE*. *SYP21* encodes a syntaxin localized to the vacuole and multivesicular bodies that physically interacts with KEU [23]. The *syp21* line carries a T-DNA insertion in the fourth exon of *SYP21* (Appendix A); homozygous plants were phenotypically wild type (Figure 6I). The *sea4 syp21* double mutants displayed small rosettes with small, highly serrated leaves (Figure 6T–U). *SYP132* encodes a syntaxin that physically interacts with KEU during cytokinesis and with SEC1B in the general secretory pathway [31]. The *syp132^T^* line carries a T-DNA insertion in the 5′ UTR of *SYP132* (Appendix A); homozygous plants displayed a wild-type phenotype (Figure 6J). The *sea4-1 syp132^T^* double mutants exhibited reduced rosette size, with small, serrated leaves. The *sea4-2 syp132^T^* double mutants showed poorly developed leaves and protuberances emerging from the apical shoot meristem (Figure 6V,W). *KNOLLE* encodes a syntaxin that plays a crucial role in cytokinesis, physically interacting with KEU at the cell plate [13,23,31]; this protein accumulated in previously studied *keu* mutants [34]. The *kn^X37−2^* mutation is a 1 kb deletion that partially removes the *KN* gene (Appendix A). Homozygous plants for this deletion were lethal, and heterozygous individuals exhibited small rosettes comprising fully expanded cotyledons and very small epinastic leaves that sometimes showed undulations (Figure 6K). The *sea4/sea4;KN/kn^X37−2^* sesquimutants exhibited a phenotype similar to that of the *KN/kn^X37−2^* heterozygotes, but their leaves were better developed, larger, and displayed undulations and serrations (Figure 6X,Y).

### 2.5. sea4 Leaves Show Increased Endoreduplication

Embryos of previously described *keu* mutants display multi-nucleate cells; these nuclei aggregate in an enlarged nucleus in seedlings [28]. To investigate whether adult plant cells exhibit the same phenotype, we conducted flow cytometry analysis of the first pair of leaves from the *sea4* mutants. As a positive control, we included *den5-1*, which carries a point mutation in *RPL7B* (encoding a ribosomal 60S subunit protein) and shows alterations in ploidy [35]. Flow cytometry revealed a decrease in populations of cells with lower ploidy levels (2C, 4C and 8C) and an increase in cell populations with higher ploidy levels (16C, 32C, 64C, and 128C; Appendix A). These altered ploidy levels were more pronounced in *sea4-1* than in *sea4-2*, which is consistent with the more severe morphological and histological phenotypes of *sea4-1* (Appendix A). An increase in ploidy usually correlates with enlarged cell size [36], but we did not observed this in the *sea4* mutants. Only a small subset of cells in the mutants were enlarged, and most were smaller than the wild type. Therefore, the increase in ploidy levels in cells of adult leaves can be more likely attributed to the aggregation of nuclei in cells unable to complete cytokinesis, as observed in previous studies on *keu* lethal seedlings.

### 2.6. Genes Related to Protein Folding, the Degradation of Misfolded Proteins, and Plant Immunity Are Upregulated in the sea4 Mutants

To determine whether the *sea4* mutations affect biological processes beyond cytokinesis, we conducted RNA-seq analysis of L*er*, *sea4-1*, and *sea4-2* rosettes collected 14 das. In *sea4-1*, we identified 3812 upregulated and 3824 downregulated genes (Appendix A). In *sea4-2*, we identified 4091 upregulated and 4034 downregulated genes (Appendix A). We classified the differentially expressed genes in *sea4-1* and *sea4-2* by GO (Appendix A) and KEGG pathway (Appendix A) analyses separately for upregulated and downregulated genes.

Many upregulated genes in the *sea4* mutants were related to protein folding and the degradation of misfolded proteins in the endoplasmic reticulum (ER). Most genes involved in the Sec-dependent protein export pathway, including all those encoding the translocation channel responsible for transporting newly synthesized proteins into the ER (protein export [KEGG:ath03060]), were upregulated in the mutants. Genes related to glycosylation, protein folding, labeling of misfolded proteins, ER-associated degradation, and the unfolded protein response (UPR) were also upregulated in the mutants (protein processing in endoplasmic reticulum [KEGG:ath04141]). Additionally, most genes encoding the 19S regulatory particle and all genes encoding the 20S proteolytic core particle of the proteasome were upregulated in the mutant, as was the gene encoding the PA200 regulatory particle, which stimulates the proteasomal hydrolysis of peptides (proteasome [KEGG:ath03050]). The glycosylation of newly synthesized peptides was also affected in the mutants, as evidenced by the upregulation of genes related to glycan biosynthesis and the oligosaccharyltransferase (OST) complex, which attaches glycan to peptides at the cytoplasmatic face of the ER membrane. Furthermore, some genes involved in modifying glycans within the ER showed increased expression in the mutants (N-glycan biosynthesis [KEGG:ath00510]; various types of N-glycan biosynthesis [KEGG:ath00513]; other types of O-glycan biosynthesis [KEGG:ath00514]).

Other upregulated genes in the *sea4* mutants are related to plant immunity (plant-pathogen interaction [KEGG:ath04626]; MAPK signaling pathway–plant [KEGG:ath04016]; immune response [GO:0006955]; regulation of defense response [GO:0031347]). Among genes involved in pathogen-associated molecular pattern (PAMP)-triggered immunity, most genes of the MAPK signaling pathways triggered by pathogen infection showed increased expression in the mutants (response to bacterium [GO:0009617]; response to fungus [GO:0009620]), leading to the production of ethylene, H_2_O_2_, and cell death (cell death [GO:0008219]). Other genes involved in MAPK signaling pathways triggered by ethylene, reactive oxygen species (ROS), salt, cold, and wounding were also upregulated in the mutants (regulation of response to stress [GO:0080134]; response to wounding [GO:0009611]). Among genes associated with effector-triggered immunity, certain genes associated with the hypersensitive response (HR) to pathogen virulence proteins were upregulated in the mutants, whereas *FIDDLEHEAD* (*FDH*), which encodes a putative 3-ketoacyl-CoA synthase and suppresses HR and defense responses via very-long-chain fatty acids, was downregulated. Genes involved in glucosinolate biosynthesis from methionine and aromatic amino acids, which are defense compounds in plants, were also upregulated in the mutant (glucosinolate biosynthesis [KEGG:ath00966]). Genes in the metabolic pathway of glutathione, which functions as an antioxidant against ROS, were differentially expressed in both *sea4* mutants (glutathione metabolism [KEGG:ath00480]); and genes in the synthesis pathway of alpha-tocopherol and alpha-tocotrienol, which are also antioxidants, were downregulated in the *sea4-1* mutants (ubiquinone and other terpenoid–quinone biosynthesis [KEGG:ath00130]).

Most genes encoding the key enzymes of the tricarboxylic acid cycle were also upregulated in the *sea4* mutants (citrate cycle [KEGG:ath00020]). Additionally, several metabolic pathways responsible for producing important metabolites in plants were upregulated in the mutants. These include the pathway that synthesizes indole-3-acetic acid from tryptophan (tryptophan metabolism [KEGG:ath00380]), the pathway that produces jasmonate from phosphatidylcholine (alpha-linolenic acid metabolism [KEGG:ath00592]), the pathway that releases abscisic acid from abscisic acid glucose ester (carotenoid biosynthesis [KEGG:ath00906]), and the pathway that produces monolignols used in lignin biosynthesis (only in *sea4-2*; phenylpropanoid biosynthesis [KEGG:ath00940]).

### 2.7. Genes Related to Photosynthesis and the Production of Various Metabolites Are Downregulated in the sea4 Mutants

Among the downregulated genes in the *sea4* mutants, many are related to photosynthetic processes (photosynthesis [GO:0015979]; photosynthesis, light reaction [GO:0019684]). Most genes encoding proteins of the F-type ATPase, photosystem I, and photosystem II (photosystem II assembly [GO:0010207]), and all genes encoding proteins of the photosynthetic electron transport chain and the light-harvesting chlorophyll complex (LHC) I and II, were downregulated in the mutants (photosynthesis [KEGG:ath00195]; photosynthesis–antenna proteins [KEGG:ath00196]; photosynthesis, light harvesting [GO:0009765]; photosynthetic electron transport chain [GO:0009767]). Furthermore, pathways responsible for the production of chlorophyll *a*/*b* and bacteriochlorophyll *a*/*b* from L-glutamate (porphyrin and chlorophyll metabolism [KEGG:ath00860]; chlorophyll biosynthetic process [GO:0015995]; chlorophyll metabolic process [GO:0015994]); as well as alpha/beta-carotene (carotenoid biosynthesis [KEGG:ath00906]), which are essential for photosynthesis, were also downregulated in the mutants. Additionally, genes encoding proteins involved in chlorophyll degradation were upregulated (porphyrin and chlorophyll metabolism [KEGG:ath00860]; porphyrin-containing compound metabolic process [GO:0006778]). In *sea4-1*, most genes of the synthesis pathways of phylloquinone, a photosystem I cofactor, and menaquinone, an electron transport chain component, were downregulated as well (ubiquinone and other terpenoid–quinone biosynthesis [KEGG:ath00130]). Most genes of the Calvin cycle (carbon fixation in photosynthetic organisms [KEGG:ath00710]), as well as the entire pathway that converts guanosine triphosphate (GTP) into flavin mononucleotide (FMN) and flavin adenine dinucleotide (FAD), were also downregulated (riboflavin metabolism [KEGG:ath00740]).

Other downregulated genes in the *sea4* mutants are related to amino acid metabolism. Genes encoding proteins involved in the conversion of glycine to glyoxylate, serine to pyruvate, and aspartate to threonine were downregulated in the mutants, although the pathway responsible for transforming serine into tryptophan was upregulated (glyoxylate and dicarboxylate metabolism [KEGG:ath00630]; glycine, serine, and threonine metabolism [KEGG:ath00260]). The pathway involved in converting aspartate to lysine was also downregulated in *sea4-2* (lysine biosynthesis [KEGG:ath00300]). Furthermore, most genes involved in the biosynthesis and degradation of valine, leucine, and isoleucine were downregulated (valine, leucine, and isoleucine biosynthesis [KEGG:ath00290]; valine, leucine, and isoleucine degradation [KEGG:ath00280]). Most genes involved in the biosynthesis of aminoacyl-tRNA, which is used by ribosomes for protein assembly, were also downregulated in the mutants (aminoacyl–tRNA biosynthesis [KEGG:ath00970]).

Lastly, genes in the pathways that interconvert alpha-D-glucose-1P, pyruvate, and acetyl-CoA were downregulated in both *sea4* mutants (glycolysis/gluconeogenesis [KEGG:ath00010]; pentose phosphate pathway [KEGG:ath00030]; pyruvate metabolism [KEGG:ath00620]). In *sea4-1*, the pathway responsible for interconverting alpha-D-glucose-1P and starch was also downregulated (starch and sucrose metabolism [KEGG:ath00500]). In addition, fatty acid metabolism was downregulated in the *sea4* mutants (fatty acid metabolism [KEGG:ath01212]). In *sea4-1*, some genes in the pathway involved in the transformation of acetyl-CoA into malonyl-[acp], as well as most genes related to the elongation pathway, were downregulated (fatty acid biosynthesis [KEGG:ath00061]). In *sea4-2*, genes involved in the biosynthesis of long-chain fatty acids, which are essential for cutin and wax biosynthesis, as well as most genes of the pathway that produces biotin from malonyl-[acp], were also downregulated in the mutant (fatty acid elongation [KEGG:ath00062]; biotin metabolism [KEGG:ath00780]).

### 2.8. The sea4 Mutants Exhibit Altered Auxin Accumulation but Normal PIN1 Distribution

In our RNA-seq analysis, both *sea4* mutants exhibited upregulation of genes related to the biosynthesis of indole-3-acetic acid. As the production and distribution of this hormone contribute to leaf shape in Arabidopsis [37], we studied its spatial distribution in the *sea4* mutants. We crossed *sea4-1* and *sea4-2* plants to the *PIN1_pro_:PIN1:GFP* transgenic line [38], which serves as a reporter for the PIN-FORMED1 (PIN1) auxin efflux carrier. Additionally, we crossed the *sea4* mutants to the *DR5rev_pro_:GFP* transgenic line [39], harboring *GFP* driven by the auxin-responsive synthetic promoter *DR5rev*.

In Col-0 leaf primordia, the *PIN1_pro_:PIN1:GFP* signal was detected in the basal region where protrusions were forming (Appendix A). In the *sea4* mutants, the spatial distribution of the *PIN1_pro_:PIN1:GFP* signal was similar but more confined to the primordial margins. The GFP signal intensity was also weaker, particularly in *sea4-1* (Appendix A). No difference in PIN1 distribution in the cells forming the primordial protrusions was observed between the *sea4* mutants and the wild type (Appendix A). The *DR5rev_pro_:GFP* signal was detected at the auxin maxima that formed in the margin protrusions and at the apex of Col-0 primordia, as well as in developing veins (Figure 7A–D). However, while the *DR5rev_pro_:GFP* signal showed the same spatial distribution in the basal region of *sea4-1* primordia as in Col-0, in the apical region of the primordia, the signal was present throughout the margin (Figure 7E–H). This change was even more pronounced in *sea4-2* primordia, with the *DR5rev_pro_:GFP* signal detected in some areas of the margin in the basal region of the primordia and in all epidermal cells of the apical region (Figure 7I–L), indicating substantial auxin accumulation. The distribution of auxin within the cells also appeared to differ. While in Col-0, the *DR5rev_pro_:GFP* signal was relatively uniform in the cytoplasm (Appendix A), in the *sea4* mutants, this signal was only detected near the cytoplasmic membrane (Appendix A).

As primary root length is reduced in the *sea4* mutants, we also studied the spatial distribution of auxin in the apical root meristem using the *PIN1_pro_:PIN1:GFP* and *DR5rev_pro_:GFP* reporters. The *PIN1_pro_:PIN1:GFP* signal localized to the center of the root in both Col-0 and the *sea4* mutants, but its intensity was weaker in the mutants, particularly in *sea4-1* (Appendix A). The distribution of PIN1 along the cytoplasmic membrane of the root cells was similar in the *sea4* mutants and Col-0. However, the cell divisions in the *sea4* mutants were poorly oriented, as evidenced by the *PIN1_pro_:PIN1:GFP* signal, which was more apparent in *sea4-1* root cells (Appendix A). The accumulation of auxin, as indicated by the *DR5rev_pro_:GFP* signal, exhibited the same spatial distribution in the *sea4* mutants as in Col-0. However, the intensity of the signal was lower in the mutants, suggesting that auxin levels were reduced at the root tip (Appendix A).

## 3. Discussion

### 3.1. sea4-1 and sea4-2 Are the First Viable Recessive Alleles of KEU

Vesicle fusion is a key process in plant cytokinesis. The accurate fusion of vesicles in the appropriate orientation ultimately determines the capacity of plant tissues to grow and differentiate correctly. The presence of the KEU protein at the division plane enables the formation of *trans*-SNARE complexes between adjacent vesicles, resulting in the formation of the cell plate [23]. Without the activity of these complexes, vesicles fuse randomly, leading to improperly oriented cytokinesis or even preventing cytokinesis altogether [28]. This is why all previously identified null *keu* mutants suffer failed cytokinesis and seedling lethality. In this study, we investigated the *sea4-1* and *sea4-2* mutants, the first known hypomorphic alleles of *KEU*, which allowed for us to explore the effects of *KEU* gene alteration in adult plants.

The T-DNA insertional lines we studied, in which the *KEU* gene is interrupted at the 17th intron (*keu21*), or 18th exon (*keu22* and *keu23*), and the *keu^MM125^* mutant, which has a 100 bp deletion in the 16th exon, are homozygous lethal. We would expect the same outcome for the *sea4-1* mutant allele, because its transcript encodes a SEA4-1 mutant protein that is even shorter. The viability of the *sea4-1* mutant allele likely results from the combination of two factors: the small percentage of mRNA transcripts that are correctly spliced at the 9th intron and its overexpression. Together, these factors result in the expression of normal mRNA in *sea4-1* at approximately 4% of the level found in wild-type Col-0. This limited expression appears to be sufficient to allow for enough vesicle fusion at the division plane during cytokinesis, allowing for the plant to complete its lifecycle.

The viability of the *sea4-2* mutant allele can be explained by the observation that the S57L substitution that it causes does not significantly alter the structure of KEU. This mutation reduces the flexibility of the protein and replaces only 1 of the 41 amino acids known to physically interact with syntaxins [32]. Altogether, the *sea4-2* mutation appears to reduce the capacity of KEU to interact with syntaxins and form *trans*-SNARE complexes between adjacent vesicles, but it does not entirely disrupt its function.

### 3.2. In the sea4 Mutants, Defects in Cytokinesis Seem to Reduce Cell Wall Integrity and Activate the Unfolded Protein Response

The viability of our hypomorphic alleles of *KEU* allowed for us to investigate their genetic interactions with related genes: those encoding its homolog SEC1B, the exocyst component SEC6, and the syntaxins SYP21, SYP132, and KN. Except for *KN*, the T-DNA insertional lines for the other genes were phenotypically wild type when homozygous. However, all their double mutant combinations with *sea4* mutations exhibited a synergistic phenotype that was quite similar: plants with fully expanded cotyledons but leaves that failed to develop correctly. These leaves produced small protuberances emerging from the shoot apical meristem and tiny, highly serrated leaves. Surprisingly, the *sea4/sea4;KN/kn^X37−2^* sesquimutants exhibited leaves that, while wavy and serrated, were more developed than *KN/kn^X37−2^* leaves.

Previous studies have indicated that the *keu* mutants display some degree of polyploidy at the lethal seedling stage [28]. We confirmed that this trait persists in adult *sea4* plants, with *sea4-1* showing more pronounced polyploidy. Higher ploidy levels typically correlate with altered levels of gene expression [40], which we also observed in the *sea4* mutants. The transcriptomic profiles of the *sea4* mutants exhibit the typical alterations in gene expression observed in mutants with the so called reduced cell wall integrity: alterations in the structural and functional stability of the cell wall, such as those caused by changes in its cellulose, pectin, or hemicellulose content, or by wall damage by external causes, which lead to modifications in cellular metabolism that regulate cell cycle progression. Some examples of mutants with reduced cell wall integrity are *irregular xylem1* (*irx1*), *irx3*, *irx5*, and *isoxaben resistant1* (*ixr1*), which carry alleles of the *CELLULOSE SYNTHASE A8* (*CESA8*), *CESA7*, *CESA4*, and *CESA3* genes, respectively; *powdery mildew-resistant5* (*pmr5*) and *pmr6*, mutated at the *PMR5* and *PMR6* genes, respectively, which encode two pectate lyases that depolymerize pectin; *de-etiolated3* (*det3*) and *alpha-xylosidase1-2* (xyl1-2), mutated at the *DET3* and *XYL1* genes, respectively, encoding proteins that modify hemicelluloses; and *walls are thin1* (*wat1*), a mutant of the *WAT1* gene, which encodes a membrane transporter and exhibits reduced secondary wall thickness [41]. Immunity is activated in reduced cell wall integrity mutants. Indeed, the *sea4* mutants showed activation of PAMP- and effector-triggered immunity, increased activity of MAPK signaling pathways, and an increase in the biosynthesis of glucosinolate defense compounds. Other effects of reduced cell wall integrity include an increase in jasmonate [42] and lignin biosynthesis [43], as also observed in the *sea4* mutants. The main pathway through which cell wall integrity seems to regulate cell division is via nitric oxide (NO) production, a process that triggers the degradation of cytokinin [44]. NO also plays a crucial role in preserving auxin sensitivity in Arabidopsis root tips [45]. Mutants affected in genes involved in NO biosynthesis, such as *NITRIC OXIDE SYNTHASE1* (*NOA1*), *NITRATE REDUCTASE1* (*NIA1*), and *NIA2*, exhibit shorter roots than the wild type. In these mutants, the distribution of PIN1 is similar to that of the wild type, but the expression of *DR5_pro_:GUS* is reduced [45]. In the *sea4* mutants, *NOA1*, *NIA1*, and *NIA2* were downregulated, resulting in a similar root phenotype. Notably, *sea4* primordia maintained the same PIN1 distribution as Col-0, but *DR5rev_pro_:GFP* expression significantly increased. This difference may be attributed to the strong upregulation of the metabolic pathway responsible for synthesizing indole-3-acetic acid in the *sea4* mutants. This upregulation could potentially compensate for the reduced auxin sensitivity caused by the deficit in NO biosynthesis.

Defects in cytokinesis have been shown to induce the UPR, which in turn increases the activity of the ER to overcome these defects [46]. One of the principal proteins initiating the UPR pathway is INCREASED ORGAN REGENERATION1 (IRE1); our RNA-seq data show that *IRE1* was upregulated in the *sea4* mutants. An increase in ER activity was also observed, with higher activities of enzymes that glycosylate proteins, transport them within the ER, and fold proteins. Genes encoding ubiquitination enzymes for misfolded proteins and the proteasome were also upregulated in the mutants, which reinforces the hypothesis of UPR activation. A possible explanation for the increase in proteolysis is that the cell detects the accumulation of vesicles and proteins in the division plane and increases the degradation of newly synthesized proteins related to cytokinesis. This explanation would also account for the reduced metabolism of some amino acids, aminoacyl-tRNA biosynthesis, and biosynthesis and elongation of fatty acids, as all these components are necessary for cytokinesis and may be in excess.

Photosynthetic light harvesting and carbon fixation were also significantly affected in the *sea4* mutants. Numerous genes encoding proteins of Light-Harvesting Complex I (LHCI) and II, photosystems I and II, the F-type ATPase, and the electron transport chain were downregulated in the mutants. Additionally, enzymes involved in the metabolic pathways that synthesize chlorophyll *a* and *b* and alpha- and beta-carotene were downregulated, as were enzymes responsible for carbon fixation in the Calvin cycle. A relationship between defects in cytokinesis or reduced cell wall integrity and the decrease in photosynthetic processes has not been previously observed. Perhaps due to their extended duration of cytokinesis, *sea4* cells remain in the M phase of the cell cycle for a longer period, leading to a reduced requirement for carbohydrate compounds. The abovementioned downregulation might also be an early sign of senesce.

### 3.3. Other Phenotypic Effects of the Altered Cytokinesis Caused by the sea4 Mutations

The phenotypes observed in the *sea4* mutants seem to be a direct consequence of their molecular alterations. The *sea4* mutations lead to defects in cytokinesis with some degree of variability, a phenomenon previously observed in other *keu* mutants [28]. This variability is visible at the tissue level, where palisade mesophyll cells vary in size, and epidermal cells cannot reach the same degree of shape complexity as in the wild type. Cells of improper size disrupt tissue organization, which is also evident in the spongy mesophyll, causing non-uniform leaf thickness and resulting in bulges. This effect can also be observed in petals, leading to premature flower opening. Disruption of internal tissue organization may also affect the boundary between dorsal and ventral leaf tissues, influencing the production and distribution of mobile signals that shape lamina growth [47]. However, it is not clear that alterations in the cell division plane are responsible for the serrations in *sea4* leaves. The Arabidopsis *lng1-1D* mutant of *TRM1* have leaves with serrated margins [48], but in the maize (*Zea mays*) *tan-1* mutant, there is no direct correlation between the orientation of cell divisions and the final shape of the leaf [5]. This is because cell divisions take place after cell elongation, which primarily determines leaf shape. Nonetheless, alterations in the cell division plane could be the cause of the reduced leaf size in the *sea4* mutants, a trait that was also observed in the *tan-1* mutant.

Although the distribution of the polar auxin transporter PIN1 remains unaltered in *sea4* leaves, the upregulation of genes in the auxin biosynthetic pathway leads to the excess accumulation of auxin, which diffuses into regions where it should not be. This ectopic auxin distribution appears to be responsible for the serrations observed in *sea4* leaves. Moreover, the presence of auxin maxima during primordia development has been linked to vascular tissue development [49,50]. The altered auxin maxima in the *sea4* mutants likely accounts for their denser and more complex vascular pattern. Reduced sensitivity to auxins in the root apical meristem via NO appears to be the reason for reduced root length. This phenomenon likely also occurs in the shoot apical meristem, explaining the observed reduction in plant height.

Premature leaf senescence is another characteristic trait of the *sea4* mutants. Genes related to plant immunity and MAPK signaling pathways are upregulated in these mutants due to a reduction in CWI. This activation triggers defense responses and, in some cases, cell death. Consequently, regions of the leaves with a high number of dead cells may form the senescent patches observed in these mutants. Defects in cytokinesis accumulate in each leaf over time and more mitotic events occur, explaining why younger leaves do not exhibit as many senescent patches as older leaves.

## 4. Material and Methods

### 4.1. Plant Materials, Growth Conditions, and Crosses

The *Arabidopsis thaliana* (L.) Heynh. wild-type accessions Landsberg *erecta* (L*er*) and Columbia-0 (Col-0), along with the *keu-21* (SALK_101874C; N661055), *keu-22* (SALK_085463; N585463), *keu-23* (SALKseq_089213; N589213), *sec1b-1* (GK-601G09; N457681; previously named *sec1b* in Karnahl et al. [31]), *sec1b-2* (GK-283F10; N427142), *syp21* (SAIL_580_C04; N875090), *syp132^T^* (SAIL_403_B09; N818666), *sec6-2* (SALK_072337C; N660954), and *sec6-3* (SALK_100970; N600970) mutants, were obtained from the Nottingham Arabidopsis Stock Center (NASC, Nottingham, UK). The *keu^MM125^* and *kn^X37−2^* lines [31] were kindly provided by Prof. Gerd Jürgens. The *sea4-1* and *sea4-2* lines were isolated in the L*er* background after EMS mutagenesis in our laboratory. Subsequently, they were backcrossed twice to L*er* [29]. Unless otherwise specified, all the mutants mentioned in this work are homozygous for the indicated mutations. Seed sterilization and sowing, plant culture, crosses, and allelism tests were conducted as previously described [29,51,52].

### 4.2. Positional Cloning and Molecular Characterization of the sea4 Mutant Alleles

Genomic DNA extraction was carried out as previously described [53]. The *sea4-1* and *sea4-2* mutations were initially mapped to a 660 kb candidate interval containing 250 genes using a mapping population of 79 F_2_ plants derived from *sea4-1* × Col-0 and *sea4-2* × Col-0 outcrosses and the primers listed in Appendix A, as previously described [53,54]. Subsequently, the complete genomes of *sea4-1* and *sea4-2* were sequenced by BGI (BGI, Shenzhen, China) using the BGISEQ platform. The raw data were analyzed using Easymap v.2 [55,56,57]. Reads were aligned to the L*er* genome [58]; the options used were based on the assumptions that the mutations under study were present in the reference genetic background, the mapping population resulted from a backcross, and the control sample was the parental line of the mutant strain. The reads from *sea4-1* were used as the control sample for *sea4-2*, and vice versa.

For plant genotyping, the wild-type *KEULE* and *sea4* mutant alleles were identified by PCR using the At1g12360-5F/R primer pair (Appendix A). The presence of T-DNA insertions in the insertional lines was confirmed by PCR, and their positions were determined via Sanger sequencing. Gene- and T-DNA-specific primers (Appendix A) were employed for these analyses.

### 4.3. Phenotypic Analysis and Morphometry

Rosettes were photographed using a Nikon SMZ1500 stereomicroscope equipped with a Nikon DXM1200F digital camera (Nikon, Tokyo, Japan). Root length was measured from photographs taken with a Canon PowerShot SX200 IS digital camera (Canon, Tokyo, Japan) using the NIS Elements AR 3.1 image analysis package (Nikon). Shoot length from the soil to the apex of the main shoot was measured in vivo using a millimeter ruler. Whole plants were photographed with a Canon PowerShot SX200 IS digital camera.

### 4.4. Differential Interference Contrast and Bright-Field Microscopy

For bright-field microscopy, all samples were cleared and mounted as previously described [59]. Micrographs of venation patterns were taken under bright field using a Nikon SMZ1500 stereomicroscope equipped with a Nikon DXM1200F digital camera and NIS Elements AR 3.1 software (Nikon). The venation pattern was drawn using Photoshop CS3 (Adobe, San José, CA, USA) on a Cintiq 18SX Interactive Pen Display screen (Wacom, Kazo, Japan) and analyzed using phenoVein software (https://quantitative-plant.org/software/phenovein, accessed on 31 January 2018; [60]). For epidermal and palisade mesophyll cell morphometry, leaves were collected and subjected to the following clearing steps: 15 min in 90% acetone, 24 h in 70% ethanol, and 24 h in 16 M chloral hydrate at room temperature. Microscopy of leaf tissues was performed using differential interference contrast optics on a Leica DMRB microscope equipped with a Nikon DXM1200F digital camera. Cell contours were manually outlined using Photoshop CS3 on a Cintiq 18SX Interactive Pen Display screen (Wacom). Cell area measurements were performed using the NIS Elements AR 3.1 image analysis package (Nikon). Transverse sections were obtained as described in Serrano-Cartagena et al. [61]). The tissue was embedded in Technovit 7100 resin (Kulzer, Hanau, Germany), and 5 μm sections were cut using a Microm HM350S microtome (Walldorf, Germany).

### 4.5. Confocal Microscopy

Confocal laser scanning microscopy images were obtained with a Leica Stellaris 8 STED confocal microscope equipped with HyD X and HyD SMD detectors, HC PL APO CS2 20×/0.75 DRY and HC PL APO CS2 40×/1.10 WATER objectives, and Leica Application Suite X software (LAS X v.4.5.0.25531; Leica, Wetzlar, Germany). Visualization of fluorescent proteins was performed on leaf primordia and primary roots mounted on glass slides in deionized water. GFP was excited at 489 nm with a white light laser (WLL). The emissions were acquired within the range of 494 nm to 583 nm, and TauSeparation was utilized to distinguish the GFP signal (2 ns) from the chlorophyll autofluorescence signal (0.12 ns). The image resolution was set to 1024 × 1024 pixels, with a speed of 600 Hz, a zoom factor of 0.75, and a line accumulation of 6. Ten optical sections, encompassing the adaxial to the abaxial epidermises of the primordia or the entire root thickness, were photographed and overlapped using LAS X software. The configuration of WLL intensity, transmitted light detector gain, and look-up table values for each photograph type are detailed in Appendix A.

### 4.6. RNA-Seq Analysis

Total RNA was extracted from the samples and subjected to massive parallel sequencing as described in Navarro-Quiles et al. [62], producing paired-end reads of 150 bp (Appendix A). Read mapping to the Arabidopsis genome (TAIR10) was performed using HISAT2 v2.0.5 [63] with default parameters, and differentially expressed genes between the *sea4* mutants and L*er* were identified by Novogene using the DESeq2 v1.20.0 R package [64]. Genes with a *p*-value < 0.05 adjusted with the Benjamini and Hochberg’s method and genes with a fold change > 1 were considered to be differentially expressed. Gene Ontology (GO; http://www.geneontology.org/, accessed on 15 March 2021) and Kyoto Encyclopedia of Genes and Genomes (KEGG; http://www.genome.jp/kegg/, accessed on 15 March 2021) pathway enrichment analyses of the differentially expressed genes were performed by Novogene using the clusterProfiler v3.8.1 R package. Significantly enriched terms were determined with an adjusted *p*-value < 0.05.

### 4.7. Ploidy Analysis

Flow cytometry analysis was performed as previously described [65]. Briefly, first-node and second-node leaves from four different rosettes were harvested 21 das and chopped with a razor blade in 500 μL of cold nuclear isolation buffer [36]. The cell suspension was filtered through a 30 μm nylon mesh, treated with RNase A (200 μg/mL) for 20 min, and stained with propidium iodide (50 μg/mL) for 40 min. The nuclear DNA content was analyzed using a FACS Canto II flow cytometer (BD Biosciences, Franklin Lakes, NJ, USA), and the data were processed using the Floreada.io web platform (https://floreada.io, accessed on 18 October 2023). Three biological replicates (15,000 counts each) were analyzed per genotype.

### 4.8. Protein Sequence Alignment

The protein sequences of Arabidopsis KEU and rat SYNTAXIN-BINDING PROTEIN1 (STXBP1) were obtained from the National Center for Biotechnology Information database (NCBI; http://www.ncbi.nlm.nih.gov/, accessed on 29 March 2021; KEU: NP_563905.1; rat STXBP1: NP_037170.1). Multiple sequence alignment was performed using Clustal Omega (EMBL-EBI, Hinxton, UK; https://www.ebi.ac.uk/Tools/msa/clustalo/, accessed on 18 March 2021; [66]).

### 4.9. Protein Structure Visualization and Analysis

The 3D structures of the full-length monomers of KEU and rat STXBP1 were obtained from the AlphaFold Protein Structure Database (https://alphafold.ebi.ac.uk/, accessed on 11 April 2023; [67,68]; KEU: AF-Q9C5X3-F1; rat STXBP1: AF-P61765-F1) and visualized using UCSF ChimeraX 1.2.5 software (https://www.rbvi.ucsf.edu/chimerax/, accessed on 11 April 2023; [69,70]). To analyze the impact of the S57L substitution in *sea4-2* on the conformational stability and dynamics of KEU, two web structure-based protein stability predictors were used: DynaMut (https://biosig.lab.uq.edu.au/dynamut/, accessed on 11 April 2023; [71]) and DynaMut2 (https://biosig.lab.uq.edu.au/dynamut2/, accessed on 11 April 2023; [72]). These predictors quantify the difference in unfolding Gibbs free energy (ΔΔG, expressed in kcal∙mol^−1^) between wild-type and mutant proteins, classifying mutations as stabilizing when ΔΔG > 0 kcal∙mol^−1^ or destabilizing when ΔΔG < 0 kcal∙mol^−1^. DynaMut provides ΔΔG results from three additional predictors: SDM [73], mCSM [74], and DUET [75]. Furthermore, it computes the difference in vibrational entropy energy (ΔΔS_Vib_, expressed in kcal∙mol^−1^∙K^−1^) between wild-type and mutant proteins using the ENCoM server [76], classifying mutations as rigidifying if ΔΔS_Vib_ < 0 kcal∙mol^−1^∙K^−1^ or flexibilizing if ΔΔS_Vib_ > 0 kcal∙mol^−1^∙K^−1^. Finally, Missense3D (http://missense3d.bc.ic.ac.uk/missense3d/, accessed on 11 April 2023; [77]) was used to predict possible damaging structural effects of the S57L substitution on KEU protein.

### 4.10. Accession Numbers

Sequence data from this article can be found at The Arabidopsis Information Resource (https://www.arabidopsis.org/, accessed on 23 January 2020) under the following accession numbers: *KEU* (At1g12360), *KN* (At1g08560), *SEC1B* (At4g12120), *SYP21* (At5g16830), *SYP132* (At5g08080), and *SEC6* (At1g71820).

## 5. Conclusions

The SM protein KEU has a known, important role in cytokinesis, where it coordinates the assembly of *trans*-SNARE complexes in vesicle fusion at the cell plate. Here, we studied the first hypomorphic, viable alleles of the *KEU* gene, *sea4-1* and *sea4-2*, which will serve as valuable resources for future research on the role of KEULE in regulating cytokinesis in postembryonic developmental stages. In *sea4-1*, a transition at the splice donor site of the ninth exon leads to mis-splicing, yielding a truncated protein. In *sea4-2*, a transition in the third exon causes a S57L substitution, which impacts an amino acid crucial for physical interactions with syntaxins and reduces the protein flexibility. In these viable mutants, cytokinesis is impaired but not abolished, allowing for the functional study of the roles that *KEU* plays in vegetative and reproductive development. Our RNA-seq study of *sea4* plants strongly suggests that their phenotypes are associated with activation of the unfolded protein response and reduction in cell wall integrity. The characteristic early leaf senescence of *sea4* plants seems to be caused by the activation of plant immunity and the increase in jasmonate biosynthesis, two typical traits of mutants with reduced cell wall integrity. Their short primary shoots and roots are also likely to be a consequence of a reduction in auxin sensitivity at the meristems due to a decrease in NO biosynthesis, another typical trait of mutants with reduced cell wall integrity. We propose that the *sea4* mutants compensate for reduced auxin sensitivity by upregulating auxin biosynthesis, which results in ectopic diffusion within leaf primordia. This leads to leaf margin serrations and a dense, complex venation pattern. The *sea4* leaf surface displays protuberances caused by internal tissue disorganization due to cytokinesis defects. Such disorganization seems to disrupt the boundaries between tissues that regulate leaf morphogenesis and may also contribute to the formation of serrated leaf margins. The *sea4* mutants also exhibit a reduction in photosynthetic light harvesting and carbon fixation, a phenomenon not previously linked to either cell wall integrity or the unfolded protein response, but that could be related to the delay in cytokinesis caused by the *sea4* mutations.

## Figures and Tables

**Figure 1 ijms-25-06667-f001:**
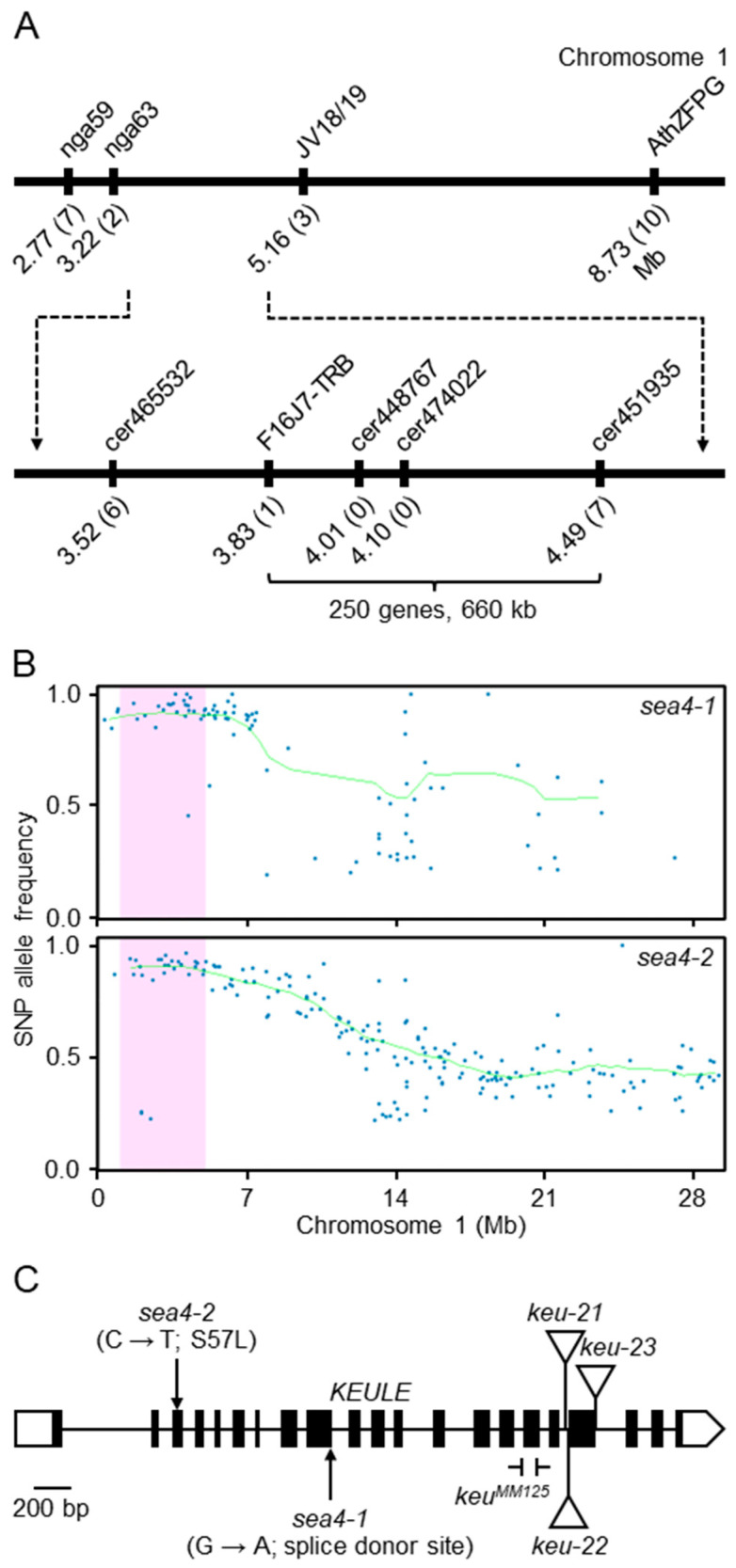
Mapping of the *sea4-1* and *sea4-2* mutations. (**A**) Analysis of a mapping population of 79 F_2_ plants derived from a *sea4-1* × Col-0 cross revealed a candidate interval of 660 kb on chromosome 1 containing 250 genes. The names and physical map positions of the molecular markers used for linkage analysis are shown. The number of recombinant chromosomes found is indicated in parentheses. (**B**) Plots showing the allele frequency of single-nucleotide polymorphisms (SNPs) versus positions in a mapping-by-sequencing analysis of the *sea4* mutants performed using Easymap v.2. SNPs are represented as blue dots, the candidate region is shaded in pink, and the average allele frequency (AF) in the test sample of SNPs used for mapping is represented by a green line. (**C**) Structure of the *KEU* gene showing the nature and positions of the *sea4* and *keu* mutations studied in this work. Boxes and lines between boxes indicate exons and introns, respectively. White boxes represent the 5′- and 3′-UTRs. Triangles indicate the T-DNA insertions in *keu-21*, *keu-22*, and *keu-23*, and the ⊣ and ⊢ symbols delimitate the deletion in *keu^MM125^*.

**Figure 2 ijms-25-06667-f002:**
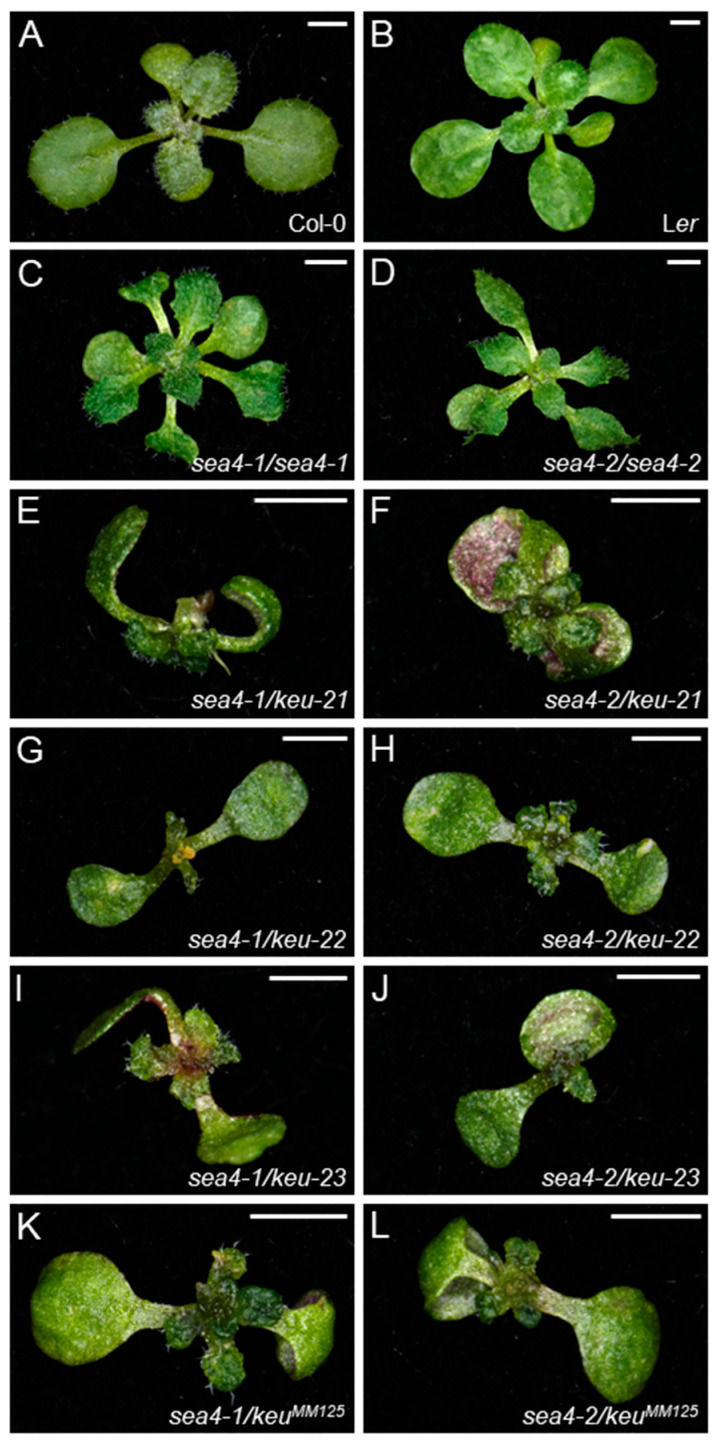
Leaf phenotypes and complementation analysis of the *sea4* and *keu* mutants studied in this work. Rosettes of wild-type (**A**) Col-0 and (**B**) L*er*, the (**C**) *sea4-1/sea4-1* and (**D**) *sea4-2/sea4-2* homozygous mutants, and the (**E**) *sea4-1/keu-21*, (**F**) *sea4-2/keu-21*, (**G**) *sea4-1/keu-22*, (**H**) *sea4-2/keu-22*, (**I**) *sea4-1/keu-23*, (**J**) *sea4-2/keu-23*, (**K**) *sea4-1/keu^MM125^*, and (**L**) *sea4-2/keu^MM125^* heterozygotes. Photographs were taken 14 das. Scale bars: 2 mm.

**Figure 3 ijms-25-06667-f003:**
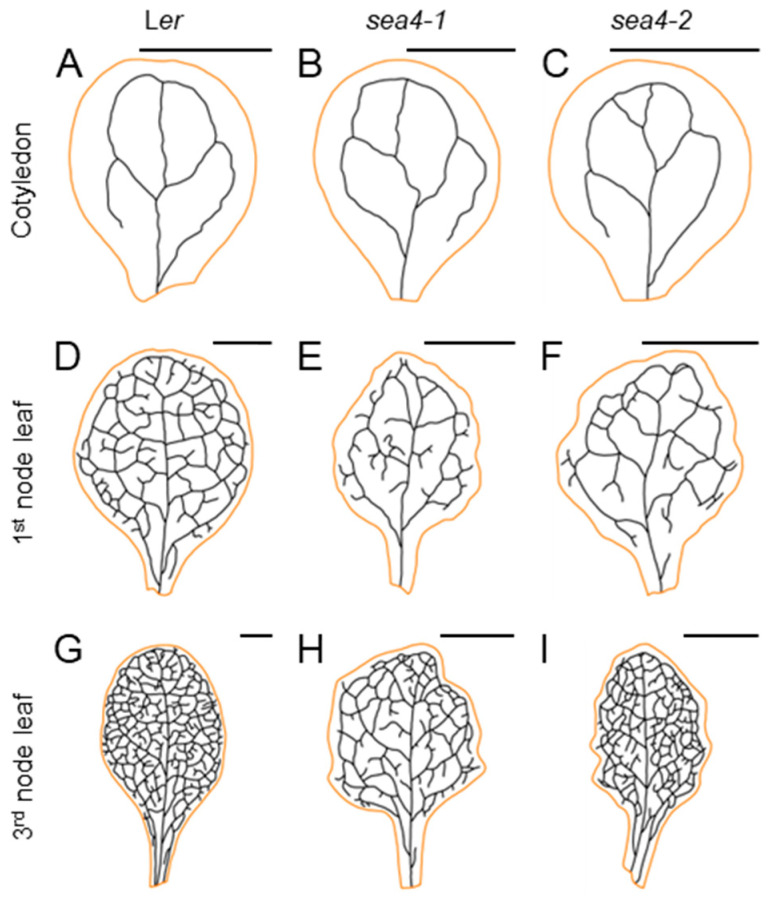
Venation patterns of *sea4-1* and *sea4-2* cotyledons and first- and third-node leaves. Representative diagrams of fully expanded (**A**–**C**) cotyledons, (**D**–**F**) first-node leaves, and (**G**–**I**) third-node leaves from (**A**,**D**,**G**) L*er*, (**B**,**E**,**H**) *sea4-1*, and (**C**,**F**,**I**) *sea4-2* plants. Margins are drawn in orange and veins in black. The line thickness does not represent the actual thickness of the veins. Cotyledons and leaves were collected 21 das. Scale bars: 2 mm.

**Figure 4 ijms-25-06667-f004:**
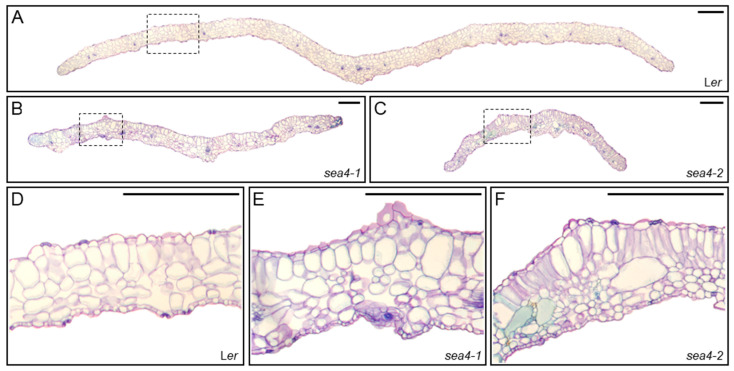
Internal structure of *sea4* third-node leaves. Transverse sections of leaves from (**A**,**D**) L*er*, (**B**,**E**) *sea4-1*, and (**C**,**F**) *sea4-2* plants stained with toluidine blue. Photographs show (**A**–**C**) a complete transverse section of the lamina from margin to margin, and (**D**–**F**) the central zone of the lamina between the primary vein and margin, which is marked with a rectangle in (**A**–**C**). Photographs were taken 21 das. Scale bars: 1 mm.

**Figure 5 ijms-25-06667-f005:**
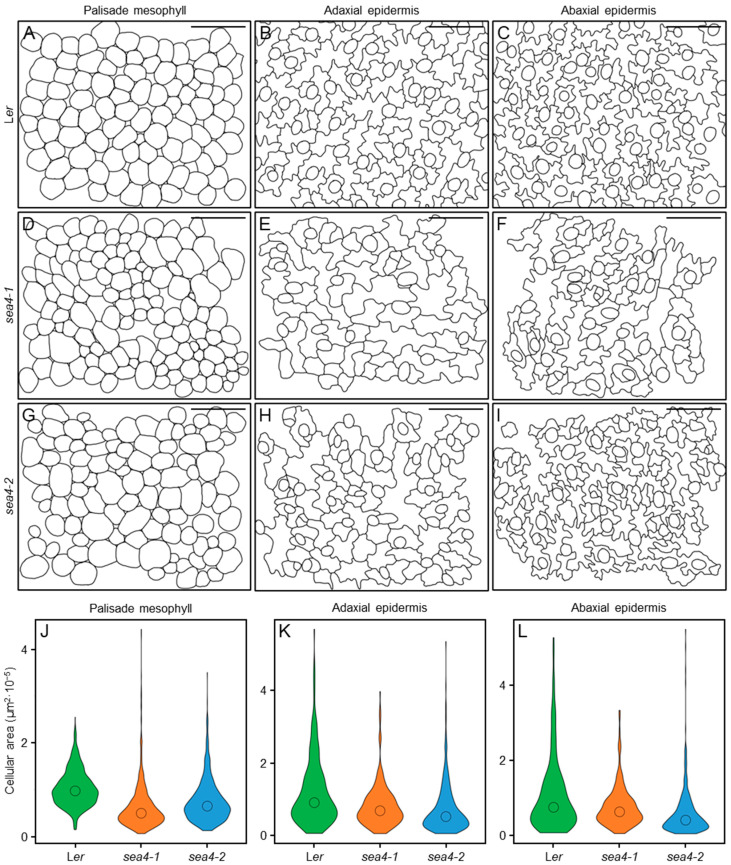
Structure of the cell layers in third-node leaves of the *sea4* mutants. (**A**–**I**) Diagrams of cells from the (**A**,**D**,**G**) palisade mesophyll, (**B**,**E**,**H**) adaxial epidermis, and (**C**,**F**,**I**) abaxial epidermis of (**A**–**C**) L*er*, (**D**–**F**) *sea4-1*, and (**G**–**I**) *sea4-2* plants. Leaves were collected 21 das. Scale bars: 20 μm. (**J**–**L**) Violin plots representing the distribution of cell sizes in the (**J**) palisade mesophyll (n = 742–982), (**K**) adaxial epidermis (n = 262–424), and (**L**) abaxial epidermis (n = 214–490). The median is represented with a circle.

**Figure 6 ijms-25-06667-f006:**
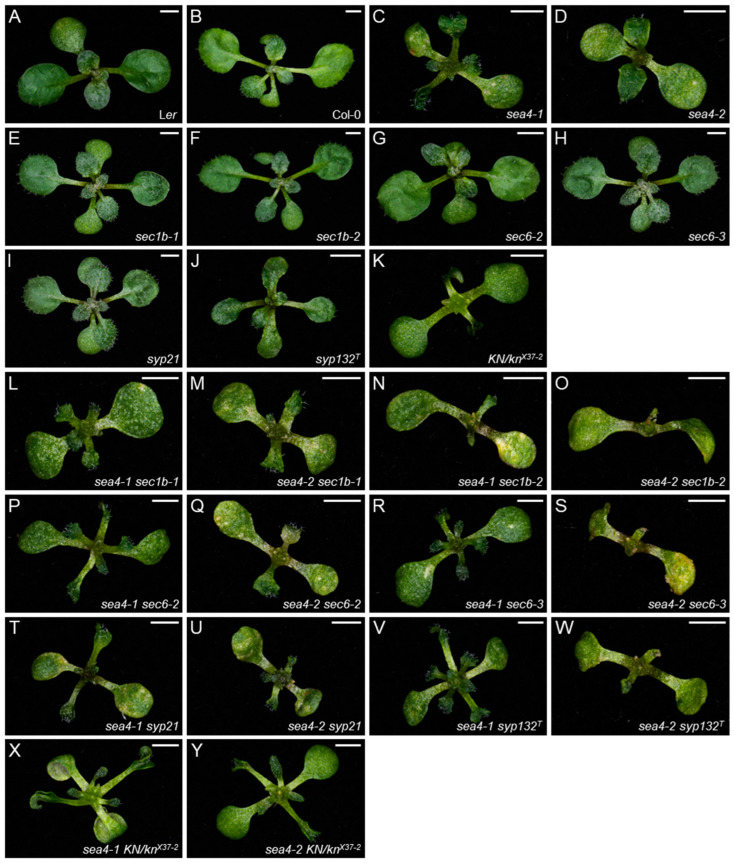
Leaf phenotypes of double mutant combinations between the *sea4* mutant alleles and mutations in genes involved in membrane fusion. Rosettes of wild-type (**A**) L*er* and (**B**) Col-0; the (**C**) *sea4-1*, (**D**) *sea4-2*, (**E**) *sec1b-1*, (**F**) *sec1b-2*, (**G**) *sec6-2*, (**H**) *sec6-3*, (**I**) *syp21*, (**J**) *syp132^T^*, and (**K**) *KN/kn^X37−2^* mutants; (**L**,**M**) *sea4 sec1b-1*, (**N**,**O**) *sea4 sec1b-2*, (**P**,**Q**) *sea4 sec6-2*, (**R**,**S**) *sea4 sec6-3*, (**T**,**U**) *sea4 syp21*, and (**V**,**W**) *sea4 syp132^T^* double mutants; and (**X**,**Y**) the *sea4 KN/kn^X37−2^* sesquimutants. Photographs were taken 14 das. Scale bars: 2 mm.

**Figure 7 ijms-25-06667-f007:**
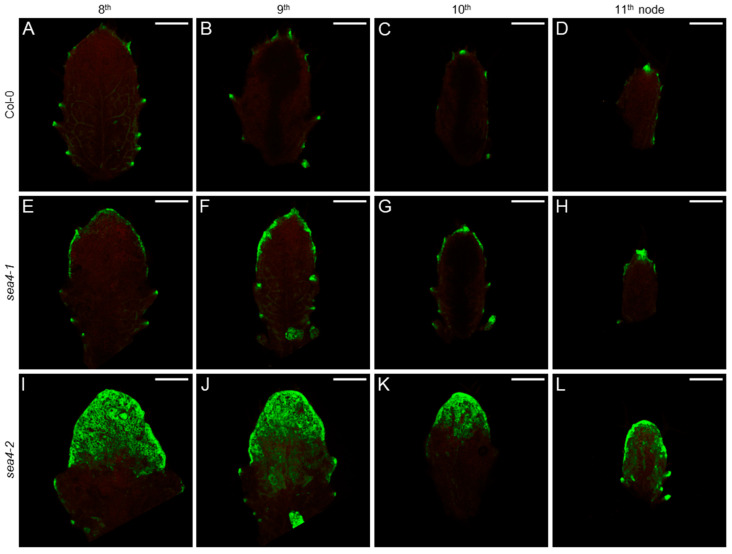
Expression pattern of the *DR5rev_pro_:GFP* reporter in leaf primordia of the *sea4* mutants (20×). The visualization of *DR5rev_pro_:GFP* (green) expression in leaf primordia from successive nodes (8th–11th) is shown for (**A**–**D**) Col-0, (**E**–**H**) *sea4-1*, and (**I**–**L**) *sea4-2* plants. Chlorophyll autofluorescence is shown in red. The primordia were collected 20 das. Scale bars: 0.2 mm.

## Data Availability

The raw data from genome resequencing and RNA-seq have been deposited in the Sequence Read Archive database (https://www.ncbi.nlm.nih.gov/sra/, accessed on 24 February 2024) under accession numbers PRJNA1079917 and PRJNA1079902, respectively.

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
