# Peer review of "Roles of the Arabidopsis KEULE Gene in Postembryonic Development"

_ijms, 2024, doi:10.3390/ijms25126667_

Round 1
Reviewer 1 Report
Comments and Suggestions for Authors
The article, authored by Ruiz-Bayón et al. is about the role of the KEU gene. Overall, the article is good in terms of content. The methods used are adequate. The article fits well with IJMS.
However, I think the authors should add some bu data to make the speculation (defects in cytokinesis through reduced cell wall integrity) more plausible.
It would be worthwhile for the authors, using DAPI or Hoechst (Nucleic Acid Stains), to show how cell division occurs in young leaves.
The authors would do well to explain what they think “reduced cell wall integrity” means.
Is it only cytokinesis that is impaired or are the chromosomes unevenly divided into daughter cells during division? Therefore, it would be desirable to color the nuclei in the cells.
Reviewer 2 Report
Comments and Suggestions for Authors
This paper presents a comprehensive study of the postembryonic developmental roles of the Arabidopsis KEU 2 gene. The research identifies the function of the Arabidopsis serrata4-1 (sea4-1) and sea4-2 mutants, which carry recessive, hypomorphic alleles of KEULE, thereby contributing to our understanding of plant genetics.
The observations are very in-depth, and there are signs that the inductive approach is feasible. However, some content needs to be strengthened to make this paper more valuable.
1. Lines 290-294, “The Arabidopsis KEU gene is expressed throughout the plant, especially in tissues undergoing division. KEU exists in soluble form in the cytoplasm or is associated with membranes during cytokinesis. KEU is involved in cytokinesis, cellular elongation, and root hair growth. KEU shares 28-30% identity with its Sec1 orthologs in mammals, Caenorhabditis elegans, and Drosophila melanogaster and 61% and 65% identity with its Arabidopsis 294 homologs SEC1A and SEC1B, respectively [13].”
Please read the reference carefully, “ Assaad, F.F.; Huet, Y.; Mayer, U.; Jürgens, G. The cytokinesis gene KEULE encodes a Sec1 protein 872 that binds the syntaxin KNOLLE. Journal of Cell Biology 2001, 152, 531-543, doi:10.1083/jcb.152.3.531.” and rewrote the content of lines 290-294.
2. In “3.3.2. The venation pattern of sea4 leaves is dense and complex”
“To observe possible alterations in the venation pattern, we decolorized cotyledons, first-node leaves, and third-node leaves of sea4-1 (n = 12) and sea4-2 (n = 13–15) plants and compared their venation patterns to Ler (n = 12–15) (Figure 3).
Could you consider including real figures, similar to Figure 4, to provide visual support for these statements and enhance the clarity of the research?
3. The style of reference needs to be revised totally.
4. Please check the grammar carefully.
Comments on the Quality of English LanguageModerate editing of English language required
